# Exacerbated Neuropathy in POLAR A and M Trials Due to Redox Interaction of PledOx-Associated Mn^2+^ and Oxaliplatin-Associated Pt^2+^

**DOI:** 10.3390/antiox12030608

**Published:** 2023-03-01

**Authors:** Jan Olof G. Karlsson, Per Jynge, Louis J. Ignarro

**Affiliations:** 1Department of Biomedical and Clinical Sciences, Division of Clinical Chemistry and Pharmacology, Linköping University, 581 83 Linköping, Sweden; 2Department of Radiology, Innlandet Trust Hospital, Gjøvik Hospital, 2819 Gjøvik, Norway; 3Department of Pharmacology, UCLA School of Medicine, 264 El Camino Drive, Beverly Hills, CA 90212, USA

**Keywords:** calmangafodipir (PledOx), chemotherapy-induced peripheral neuropathy, clinical phase III trials, manganese, oxaliplatin, drug interaction, POLAR A and M trials, platinum, redox active cations

## Abstract

Disappointing results from the POLAR A and M phase III trials involving colorectal cancer patients on chemotherapy with FOLFOX6 in curative (A) and palliative (M) settings have been reported by the principal investigators and the sponsor (PledPharma AB/Egetis Therapeutics AB). FOLFOX6, oxaliplatin in combination with 5-fluorouracil (5-FU), possesses superior tumoricidal activity in comparison to 5-FU alone, but suffers seriously from dose-limiting platinum-associated Chemotherapy-Induced Peripheral Neuropathy (CIPN). The aim of the POLAR trials was to demonstrate that PledOx [calmangafodipir; Ca_4_Mn(DPDP)_5_] reduced the incidence of persistent CIPN from 40% to 20%. However, this assumption was based on “explorative” data in the preceding PLIANT phase II trial, which did not mirror the expected incidence of unwanted toxicity in placebo patients. In POLAR A and M, the assessment of PledOx efficacy was conducted in patients that received at least six cycles of FOLFOX6, enabling analyses of efficacy in 239 A and 88 M patients. Instead of a hypothesized improvement from 40% to 20% incidence of persistent CIPN in the PledOx group, i.e., a 50% improvement, the real outcome was the opposite, i.e., an about 50% worsening in this bothersome toxicity. Mechanisms that may explain the disastrous outcome, with a statistically significant number of patients being seriously injured after having received PledOx, indicate interactions between two redox active metal cations, Pt^2+^ (oxaliplatin) and Mn^2+^ (PledOx). A far from surprising causal relationship that escaped prior detection by the study group and the sponsor. Most importantly, recently published data (ref 1) unequivocally indicate that the PLIANT study was not suited to base clinical phase III studies on. In conclusion, the POLAR and PLIANT trials show that PledOx and related manganese-containing compounds are unsuited for co-treatment with platinum-containing compounds. For use as a therapeutic adjunct in rescue treatment, like in ischemia-reperfusion of the heart or other organs, or in acetaminophen (paracetamol)-associated liver failure, there is little or nothing speaking against the use of PledOx or other PLED compounds. However, this must be thoroughly documented in more carefully designed clinical trials.

## 1. POLAR A and M Clinical Phase III Trials

The POLAR A phase III trial was conducted in colorectal cancer patients going through curative FOLFOX6 chemotherapy with the intention to prove that co-treatment with PledOx [calmangafodipir; Ca_4_Mn(DPDP)_5_] lowers the incidence of persistent oxaliplatin-associated chemotherapy-induced sensory neuropathy (CIPN) by 50% [1]. The POLAR M phase III trial was more or less an identical trial but included palliative patients instead of curative patients. On 6 April 2020, these trials were terminated due to “allergic” hypersensitivity reactions in PledOx-treated patients, first reported by Qvortrup et al., 2021 [2] to equal twelve in the PledOx group and later reported by Pfeiffer et al., 2022 [1] to equal nine. At that timepoint, the POLAR M trial was already put on hold since 1 March, 2021 by the FDA and the responsible regulatory authority in France because of suspected manganese-related toxicity in the PledOx group. At closure, the POLAR A study was fully recruited and enabled efficacy assessments of 5 µmol/kg PledOx and placebo in 120 and 119 patients, respectively (ClinicalTrials.gov; NCT04034355). The recruitment in the POLAR M trial laid behind the POLAR A and enabled efficacy assessment of 2 µmol/kg PledOx, 5 µmol/kg PledOx and placebo in 31, 27 and 25 patients, respectively (ClinicalTrials.gov; NCT03654729).

## 2. Therapeutic Efficacy of PLED Compounds and Importance of Scrutinizing What Went Wrong in the POLAR Trials

The MRI contrast agent mangafodipir (MnDPDP; Teslascan) and derivatives thereof, known as PLED (diPyridoxyl EthylDiamine) compounds (Figure 1), have shown therapeutic efficacy in a wide range of serious pathological condition preclinically as well as clinically [3,4,5,6,7,8,9,10,11,12,13,14,15,16,17,18,19].

However, highly negative results were reported from the POLAR A and M phase III clinical trials. Instead of an anticipated 50% decrease in the incidence of persistent oxaliplatin-related CIPN in cancer patients treated with PledOx, a diametrically opposite outcome was obtained, namely an about 50% exacerbation of this bothersome toxicity in the POLAR A trial [1]. The incidence of CIPN in PledOx-treated patients was lower in the POLAR M trial but was still higher than in the placebo group.

As founders of PledPharma, a company founded to develop PLED substances into approved rescue drugs for treatment of life-threatening conditions, such as acute myocardial infarction, acetaminophen (paracetamol)-induced liver failure and fibril neutropenia, we find it important to discuss what might have gone wrong in the above-described trials, and what kind of mechanisms may explain the disastrous outcome, where a statistically significant number of patients have been seriously injured after having received PledOx. Such discussions may hopefully help to avoid similar negative outcomes in future clinical trials.

Although, the principal investigators (IPs) have entered an agreement with the sponsor that restricts the PIs’ rights to discuss or publish trial results (see ClinicalTrials.gov; NCT04034355 and NCT NCT03654729), we nevertheless look forward to receiving their comments.

## 3. Reported Results from the POLAR A and M Trials

The core reported results by Pfeiffer et al., 2022 [1], confirmed those previously disclosed by Qvortrup et al., 2021 [2], showing that PledOx caused a 37% (*p* = 0.0445) increase in the incidence of persistent CIPN in A and M pooled patients. The exacerbating effect was even larger in the POLAR A study alone where PledOx increased the incidence by 52% (*p* = 0.028) [1]. In the preceding power analyses, both studies were estimated to display a decrease in the incidence from 40% to 20% in PledOx treated patients, i.e., a 50% improvement [1]. However, in the POLAR A study, PledOx in fact increased the incidence of persistent CIPN from roughly 40% to 60% (see ClinicalTrials.gov; NCT04034355), i.e., a 50% exacerbation. To characterize such an outcome as “failure to meet primary endpoint” as the authors do [1,2] is misleading.

Furthermore, seeking a genetic explanation for the failure, the authors refer to data supported by *p*-values of about 0.5. However, the most obvious explanation of the failure was the premature decision to advance PledOx into phase III based on the PLIANT phase II study [20].

## 4. What Caused the POLAR Failure?

A common aim of a clinical phase III trial is to confirm the main result obtained in a preceding phase II trial. This result defines the primary endpoint and forms the statistical prerequisites for the subsequent phase III study, from which a so-called power analysis is done to determine a good sample size for a particular effect. Such analysis demands that a certain effect of the test drug has been obtained, compared to a placebo drug. However, the overriding problem with the PLIANT trial was the exceedingly small number of oxaliplatin-related adverse events (AEs) in placebo patients, including oxaliplatin-induced peripheral neuropathy [20]. Neither the original primary endpoint, i.e., neutropenia, nor any other endpoint was reached in the PLIANT trial.

It is essential that a clinical trial aimed to test whether a particular treatment reduces AEs, in fact roughly mirrors the expected frequency of AEs in the placebo group—which PLIANT did not. Karlsson and Jynge [21] raised serious criticism in a Letter to the Editor of *Acta Oncologica* (where the PLIANT study was published) against the decision to advance PledOx into phase III based on the PLIANT trial. Karlsson and Jynge characterized the decision as a “hazardous” one. However, three of the authors of PLIANT publication maintained in their reply that the decision was based on “trustworthy” data [22].

The selected primary endpoint in the POLAR studies, i.e., patient-reported persistent CIPN, according to the FACT/GOG-NTX-13 subscale questionnaire, targeting numbness, tingling, or discomfort in hands and/or feet. This questionnaire was not included in the PLIANT trial but was based on another patient-reported questionnaire, according to the Leonard scale, targeting “mean sensory score of the average sum of tingling, numbness and burning pain to cold in hands and feet”, as presented in Figure 3c in the PLIANT publication [20]. Although this figure seemingly displays some real neurotoxic effects of oxaliplatin, presenting the results as the average sum of several variables, where the maximum effect for each of them equals 10 on a 0–10 scale, artificially enlarges the mean effect. In the POLAR trials, the four used variables were instead analysed out from the criterion “in at least 1 of the first 4 items”. The PLIANT results were furthermore based on pooled data from three different dose groups, which of course, further belittles the value of the results. Furthermore, presenting the results as mean ± SEM (SEM = SD/$\sqrt{n})$ with sample sizes around 30 gives much smaller SEM than the true variabilities, SD, within the samples. Multiplying SEM with 1.96 gives an approximate 95% confidence interval, which in turn, reveals a huge overlap in these intervals, as presented in Figure 3c in Glimelius et al., 2017 [20]. The true incidence of persistent CIPN still appears exceedingly small in both the placebo and PledOx groups in the PLIANT trial.

Furthermore, physician-judged CIPN-incidence under eight cycles of FOLFOX chemotherapy in both placebo patients and PledOx patients in the PLIANT trial, as shown in Figure 2, Glimelius et al., 2017 [20], was also exceedingly small. Taken in consideration that the original primary endpoint of the PLIANT actually was grade 3/4 neutropenia, it should also be noted that this incidence was similarly low, 12% instead of the expected 40%, and more in line with what you expect from 5-FU alone. Hence, the profile of AEs corresponded much better to 5-FU alone than that of the combination of oxaliplatin and 5-FU, as noted by Karlsson and Jynge already in 2017 [21].

PLIANT authors presented at the MASCC meeting in Copenhagen, 2015, and at the ASCO meeting in Chicago, 2016, an objective response rate (ORR) of 27% in the placebo group of the PLIANT trial. The reported ORR of 27% in the placebo group is identical to what is expected from 5-FU alone (cf. 18). Addition of oxaliplatin to 5-FU has increased the ORR from about 27% to 45%. Karlsson and Jynge notified the sponsor on the huge discrepancy between the expected tumoricidal efficacy and of that obtained in the PLIANT trial. Before the PLIANT trial was published in *Acta Oncologica* [20], Karlsson was informed by Glimelius that the ORR was incorrect due to a miscalculation done by the clinical contract research organisation, PCG Clinical Services AB in Uppsala. After recalculation, the ORR was changed from 27% to 43% in the placebo group, a rather expected figure for FOLFOX6, but the progression-free survival (PFS) was still no longer than 7 months and the incidence of AEs remained exceedingly low [21].

There was nothing, whatsoever, in the PLIANT trial supporting the decision to advance PledOx into phase III. Neither was there evidence in the PLIANT trial supporting the power analysis of the POLAR trials, anticipating a 40% incidence of persistent CIPN in the placebo groups and a 20% in the PledOx group.

## 5. Redox Interaction between Mn^2+^-Containing PledOx and Pt^2+^-Containing Oxaliplatin

In vivo mixing of two metal complexes (oxaliplatin and PledOx) with inherent redox properties may lead to devastating drug interactions. In a recent opinion article in *Antioxidants* [23], we discussed possible mechanisms behind the POLAR failure, suggesting that it is explained by intravenous administrations of PledOx and oxaliplatin being too close in time and, thereby, causing unfavorable redox interactions between Mn^2+^ and Pt^2+^. In that discussion, we probably proceeded from an incorrect assumption of an acute interaction between Pt^2+^ and Mn^2+^. A more plausible explanation may be a persistent interaction, where these redox active cations co-accumulate in the dorsal root ganglion (DRG) and where Pt^2+^ oxidizes Mn^2+^ into Mn^4+^, and where Mn^4+^ drives the devastating protein nitration, as schematically shown in Figure 2.

## 6. Involvement of Manganese in Mitochondrial Protein Nitration

During high cellular oxidative stress, nitric oxide (NO) readily reacts with O_2_^−^ and this forms peroxynitrite (ONOO^−^), which is in the presence of oxometal complexes, e.g., O = Fe^4+^X and O = Mn^4+^X [18,24]. These two species are able to oxidize and nitrate Tyr-residues, including site specific nitration of human Tyr34, located about 5Å from the active site of the mitochondrial manganese superoxide dismutase [24]. These are conditions that irreversibly inactivate this critical enzyme. Furthermore, Tyr74 of cytochrome c is another mitochondrial target for Mn^4+^-driven nitration, triggering a conformational change, resulting in an alternative conformation lacking its normal electron transport capacity. Altogether, this will result in a critically exacerbated situation. Protein tyrosine nitration has for many years been known to occur in several pathologies, such as cardiovascular disease, neurodegeneration, inflammation, and cancer [25].

## 7. Plausible Mechanism behind PledOx-Induced CIPN Exacerbation

The clinical trial by Coriat et al., 2014 [16] is of particular interest when it comes to persistent CIPN, suggesting that co-treatment with MnDPDP (Teslascan) not only protects the patients from CIPN, but in fact also cured ongoing oxaliplatin-related CIPN. We have in a previous *Antioxidants* publication [23] discussed the importance of distancing the administrations of PledOx and oxaliplatin in order to avoid negative redox interactions. In the Coriat trial, MnDPDP was administered immediately upon the oxaliplatin infusion, which may indicate that the distance between administrations does not really matter. However, the CIPN exacerbation in the POLAR trials was observed 9 months after the start of chemotherapy, and we are not aware of such long follow-ups in the Coriat trial. Furthermore, asserting that such huge exacerbation should be caused by an acute and transient redox interaction between platinum and manganese is less plausible. A more plausible explanation seems to be co-accumulation and retention of both these metals in the DRG. Similar arguments are also viable when it comes the 6 mM content of the antioxidant ascorbic acid in the ready-to-use formulation of MnDPDP (Teslascan), i.e., it will not have any crucial effect on the final outcome.

There are good reasons to anticipate that intravenously administrated organic platinum as well as manganese will accumulate in nerve tissue, particularly in the DRG (Figure 2), located outside the blood-brain barrier and lacking a draining lymph system and cerebrospinal fluid. This makes potentially dangerous substances, such as chemotherapy drugs and toxic metals, to accumulate in the peripheral nerve system and causes oxidative stress and detrimental nerve injuries (cf. 18). Furthermore, oxaliplatin is a small (Mw 397) and highly lipophilic compound with a distribution volume of almost 600 L [26]. Oxaliplatin will therefore readily get intracellular access to the DRG cells. Similarly, the MnDPDP metabolite MnPLED is a small and lipophilic compound that is expected to readily get intracellular access to DRG neurons.

Platinum(II), with its high reduction potential, oxidizes Mn^2+^ into highly toxic Mn^4+^, which in turn drives nitration of tyrosine residues as described above, leaving the mitochondrion without antioxidant protection. This is of course extremely critical for neurons, such as the DRG neurons. Similar negative interactions may also explain the “allergic” hypersensitivity reactions and the manganese toxicity seen in the POLAR trials, as notified by us in a previous paper in *Antioxidants* [23].

Manganese is an essential metal present in nerve tissue, e.g., in the active site of the mitochondrial MnSOD enzyme, where it disarms devastating superoxide radicals (O_2_^−^) [3]. Intravenously administered Pt^2+^ (associated to oxaliplatin) is taken up by DRG cells, either via metal transporters or through passive diffusion. This may in turn drive oxidation of endogenous Mn^2+^ into Mn^4+^, resulting in an irreversible inactivation of both mitochondrial MnSOD and cytochrome c (Figure 2). Interestingly, such devastating reactions may in fact be a plausible mechanism behind oxaliplatin-associated persistent CIPN presence, driven by oxidation of endogenous Mn^2+^/Mn^3+^. Interestingly, protein nitration is implicated in the development of diabetic peripheral neuropathy [27].

Importantly, the new publication by Pfeiffer et al., 2022 [1] discloses the true timepoint for administration of PledOx, which previously has been defined as “on top of modified FOLFOX6” (2; ClinicalTrials.gov; NCT03654729 and NCT04034355). However, as the POLAR study reported a closely similar administration regimen as that used in the preceding phase II PLIANT study, i.e., an intravenous PledOx infusion 10 min ahead of oxaliplatin, it clearly indicates that the literally negative effect of PledOx should have been easily detected in a properly executed and monitored phase II study. That is, a study displaying the expected incidence of oxaliplatin-induced CIPN in placebo patients, similar to the CIPN incidence displayed in the POLAR A and M studies.

## 8. An Alternative and Manganese-Independent Possibility to Treat Platinum-Associated CIPN

As noted above, the clinical trial by Coriat and co-workers 2014 [16] suggested that co-treatment with MnDPDP not only protected the patients from CIPN, but in fact also cured ongoing oxaliplatin-related CIPN. The authors maintained that these effects were due to the MnSOD-mimetic activity of MnDPDP. However, this explanation is questionable from a pharmacokinetic perspective, where the MnSOD mimetic activity is only expected to last a couple of hours. Another but until recently undisclosed possibility is that Pt^2+^ has a high enough formation constant (^10^logK_ML_) to outcompete Mn^2+^/Ca^2+^/Zn^2+^ for binding to DPDP, or its dephosphorylated metabolite PLED, and transforms toxic Pt^2+^ into a non-toxic complex, which can be excreted from the body by a process known as chelation therapy. As opposed to the MnSOD mimetic alternative, chelation therapy may solve the underlying problem, namely retention of Pt^2+^ in the DRG. Electron paramagnetic resonance (EPR)-guided competition experiments between MnDPDP (^10^logK_ML_ ≈ 15) and K_2_PtCl_4_, and between MnDPDP and ZnCl_2_ (^10^logK_ML_ ≈ 19), respectively, from which an estimate of the ^10^logK_ML_ of PtDPDP was obtained. This estimate clearly suggested that DPDP has high enough affinity for acting as a chelation drug [28].

The study by Canta et al., 2020 [29], utilizing a mouse model of oxaliplatin-induced peripheral sensory neuropathy, is of central interest. In this study, histopathological findings, after eight weeks of follow up, demonstrated significant neuroprotective efficacy of PledOx. When it came to the mechanistic part of the conclusion, it was expressed as “the current study lends mechanistic support and insights to the findings that the iron chelator and SOD-mimetic calmangafodipir and the related mangafodipir have demonstrated clinical efficacy against OHP-induced CIPN”. Although, both iron chelation and MnSOD mimetic activity in the first glance may appear trustworthy. Platinum-associated CIPN is in our eyes caused by long retention of Pt^2+^ in the body. Significantly increased levels in the body can be detected 10 to 20 years after chemotherapy, causing a long-lasting oxidative and nitrosative stress and chronic toxicity. From a pharmacokinetic perspective, a few hours MnSOD mimetic activity in conjunction with chemotherapy is probably of negligible help [28]. Similar arguments could also be used against iron-chelation. Interestingly, an Electron Paramagnetic Resonance (EPR) study has demonstrated that Pt^2+^ is binding to the DPDP-chelator with high affinity [28]. A more plausible mechanism than those described by Canta et al., is that DPDP (or its metabolite PLED) binds Pt^2+^ and forms a low molecular weight complex that can be excreted through the kidneys.

According to available information, the Canta et al. study [29] also included paclitaxel, a chemotherapy drug causing similar peripheral neuropathy as that of oxaliplatin. However, the etiology differs between these two forms of neuropathy; in the case of oxaliplatin, it is caused by platinum retention, and in the case of paclitaxel, it is caused by binding of paclitaxel to microtubules and a consequent inhibition of axonal transport. Since the result of paclitaxel is not reported, we presume that the result was negative, i.e., PledOx did not offer any neuroprotective efficacy against paclitaxel. The presumed negative result in turn suggests that the neuroprotective efficacy of PledOx, as that seen in mice, is a property independent of Mn^2+^, through a so-called chelation therapeutic mechanism of action.

## 9. Conclusions

A costly lesson to be learned from POLAR trials, for both participating patients and shareholders, is that PledOx should not be used in combination with platinum-containing drugs, such as oxaliplatin and cisplatin. This should have been obvious already during a well conducted and monitored phase II trial.

## Figures and Tables

**Figure 1 antioxidants-12-00608-f001:**
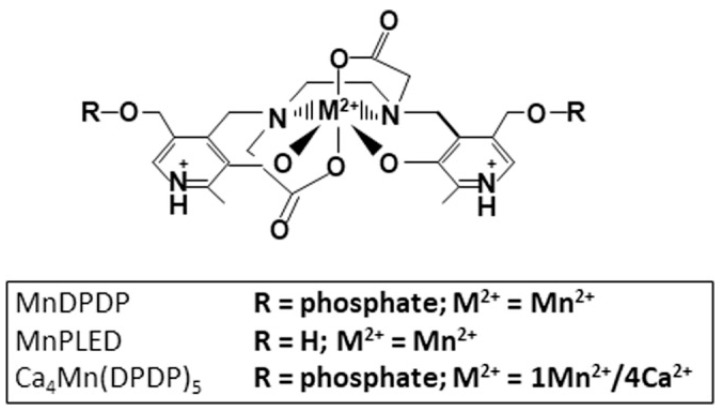
Chemical structure of MnDPDP (manganese dipyridoxyl diphosphate; generic name mangafodipir), MnPLED (manganese dipyridoxyl ethyldiamine), and Ca_4_Mn(DPDP)_5_ [tetracalcium monomanganese penta(dipyridoxyl diphosphate); calmangafodipir].

**Figure 2 antioxidants-12-00608-f002:**
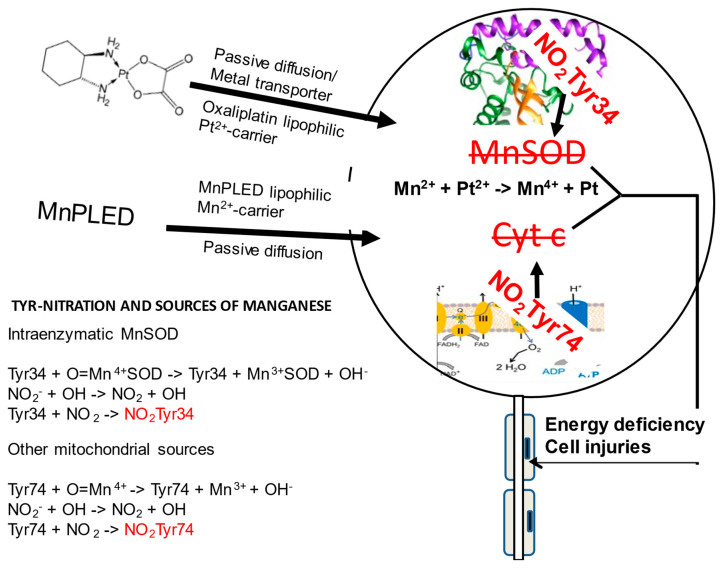
Schematic illustration of a DRG neuron showing two tyrosine nitration reactions taking place during high oxidative stress, relevant to co-administration of manganese-containing PledOx and platinum-containing oxaliplatin. High oxidative stress due the presence of Pt^2+^ causes oxidation of Mn^2+^/Mn^3+^ to Mn^4+^ and simultaneously a reaction between superoxide (O_2_^−^) and nitric oxide (NO) forming highly toxic peroxynitrite (ONOO^−^). In presence of Mn^4+^, ONOO^−^ nitrates two mitochondrial tyrosine residues, Tyr34 in the hMnSOD and Tyr74 in the cytochrome c, resulting in irreversible enzyme inactivation and disturbed electron transport, as illustrated to the right. DRG influx of lipophilic oxaliplatin and the MnDPDP metabolite MnPLED is illustrated to the left, and below that illustration, manganese sources together with tyrosine nitration chemistry are shown.

## Data Availability

Not applicable.

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
