# Peer review of "Exacerbated Neuropathy in POLAR A and M Trials Due to Redox Interaction of PledOx-Associated Mn2+ and Oxaliplatin-Associated Pt2+"

_antioxidants, 2023, doi:10.3390/antiox12030608_

Round 1

Reviewer 1 Report

antioxidants-2145126-peer-review

This manuscript is opinion on recently published POLAR A and M clinical phase III studies.I have main concerns on the critical communication tone. Manuscript seems to be of a critical review not the communication.

I agree, that clinical trials should be scientifically sound and properly planned and conducted. However, I’d like to suggest the authors more concentrating on the opinion than open critticism.

Author Response

The authors have carefully revised the manuscript with aim to tone down the critical language. The title has in addition soften   

Reviewer 2 Report

The subject of the short communication presented for a review is of certain interest and importance. I would suggest that authors cite and discuss more literature sources in order better support of the conclusion that the disastrous outcome of the POLAR A and M clinical phase III studies could be avoided.

Author Response

We agree that there is a need of a broader publication. However, in the first run it is essential to scrutinize the meaning of the PIs statement that the POLAR studies "did not meet the primary endpoint". Not at least in respect to the participating cancer patients, whose expectation were quite different from the actual outcome of the POLAR studies.  Already on December 15, 2020, the sponsor reported in a press release that the study "did not
meet the efficacy endpoint" but without giving any detailed information. Two years later we presume the time has come scrutinize the meaning statement, from the participating patient viewpoint. The true meaning is that 20 more patients in the A (adjuvant) study were afflicted by persistent (chronic) neuropathy in the PledOx group compared to the placebo group, corresponding to an incidence of 57% compared to 40% in the placebo patients. Unfortunately, it is not possible to differentiate these 20 patients that had got chronic CIPN as result of a combined neurotoxic effect of oxaliplatin plus PledOx from those 40 that have received oxaliplatin alone. Taken in consideration that the 5-year survival is about 70% in this group of patients, a considerable amount of the participating patients that received PledOx are alive and suffer today from PledOx-related chronicneuropathy. As the exacerbation of chronic neuropathy was obviously caused by a premature phase III advance of PledOx, the disaster could have been avoided. The exacerbation was somewhat less in the M-part of the POLAR study but still significant.

Reviewer 3 Report

It would be better to tone down the language of the title.

The title would be better as 

Possible reason for lack of efficacy of PledOx in the POLAR A and M phase III studies

It would be usefu

While the oxaliplatin-induced peripheral sensory neuropathy mouse study of Canta et al. 2020 showed significant neuroprotective efficacy of PledOx, it is hard to translate the results to human studies owing to the stark differences in the metabolism between mouse and human (as the authors themselves point out. 

Author Response

The suggested title by the reviewer, does in our opinion not fully cover the content of our Communication. However, we hope that the new title may suit. 

Round 2

Reviewer 1 Report

The article has been improved and modified as per the suggestions.

Reviewer 2 Report

I am satisfied with the improvements that the authors have made to the manuscript following the reviewer's comments and recommendations